# Budgeted Training:
# Rethinking Deep Neural Network Training Under Resource Constraints

**Mengtian Li**[†]
Carnegie Mellon University
mtli@cs.cmu.edu

**Ersin Yumer**[†]
Uber ATG
meyumer@gmail.com

**Deva Ramanan**
CMU & Argo AI
deva@cs.cmu.edu

## Abstract

In most practical settings and theoretical analyses, one assumes that a model can be trained until convergence. However, the growing complexity of machine learning datasets and models may violate such assumptions. Indeed, current approaches for hyper-parameter tuning and neural architecture search tend to be limited by practical resource constraints. Therefore, we introduce a formal setting for studying training under the non-asymptotic, resource-constrained regime, i.e., budgeted training. We analyze the following problem: "given a dataset, algorithm, and *fixed* resource budget, what is the best achievable performance?" We focus on the number of optimization iterations as the representative resource. Under such a setting, we show that it is critical to adjust the learning rate schedule according to the given budget. Among budget-aware learning schedules, we find simple linear decay to be both robust and high-performing. We support our claim through extensive experiments with state-of-the-art models on ImageNet (image classification), Kinetics (video classification), MS COCO (object detection and instance segmentation), and Cityscapes (semantic segmentation). We also analyze our results and find that the key to a good schedule is budgeted convergence, a phenomenon whereby the gradient vanishes at the end of each allowed budget. We also revisit existing approaches for fast convergence and show that budget-aware learning schedules readily outperform such approaches under (the practical but under-explored) budgeted training setting.

## 1 Introduction

Deep neural networks have made an undeniable impact in advancing the state-of-the-art for many machine learning tasks. Improvements have been particularly transformative in computer vision (Huang et al., 2017b; He et al., 2017). Much of these performance improvements were enabled by an ever-increasing amount of labeled visual data (Russakovsky et al., 2015; Kuznetsova et al., 2018) and innovations in training architectures (Krizhevsky et al., 2012; He et al., 2016).

However, as training datasets continue to grow in size, we argue that an additional limiting factor is that of resource constraints for training. Conservative prognostications of dataset sizes – particularly for practical endeavors such as self-driving cars (Bojarski et al., 2016), assistive medical robots (Taylor et al., 2008), and medical analysis (Fatima & Pasha, 2017) – suggest one will train on datasets orders of magnitude larger than those that are publicly available today. Such planning efforts will become more and more crucial, because *in the limit, it might not even be practical to visit every training example before running out of resources* (Bottou, 1998; Rai et al., 2009).

We note that resource-constrained training already is *implicitly* widespread, as the vast majority of practitioners have access to limited compute. This is particularly true for those pursuing research directions that require a massive number of training runs, such as hyper-parameter tuning (Li et al., 2017) and neural architecture search (Zoph & Le, 2017; Cao et al., 2019; Liu et al., 2019).

---

[†]Work done while at Argo AI.

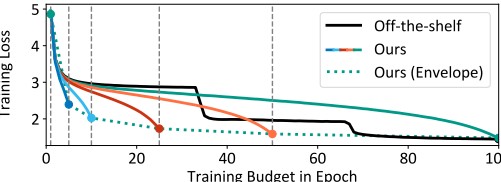 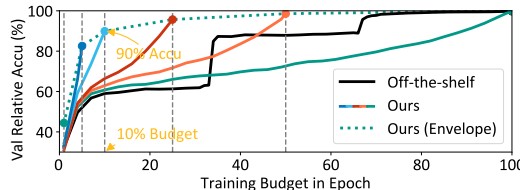

Figure 1: We formalize the problem of *budgeted training*, in which one maximizes performance subject to a fixed training budget. We find that a simple and effective solution is to adjust the learning rate schedule accordingly and anneal it to 0 at the end of the training budget. This significantly outperforms off-the-shelf schedules, particularly for small budgets. This plot shows several training schemes (solid curves) for ResNet-18 on ImageNet. The vertical axis in the right plot is normalized by the validation accuracy achieved by the full budget training. The dotted green curve indicates an efficient way of trading off computation with performance.

Instead of asking "what is the best performance one can achieve given this data and algorithm?", which has been the primary focus in the field so far, we decorate this question with *budgeted training* constraints as follows: "what is the best performance one can achieve given this data and algorithm within the allowed budget?". Here, the *allowed budget* refers to a limitation on the total time, compute, or cost spent on training. More specifically, we focus on limiting the number of iterations. This allows us to abstract out the specific constraint without loss of generality since any one of the aforementioned constraints could be converted to a finite iteration limit. We make the underlying assumption that the network architecture is constant throughout training, though it may be interesting to entertain changes in architecture during training (Rusu et al., 2016; Wang et al., 2017).

Much of the theoretical analysis of optimization algorithms focuses on asymptotic convergence and optimality (Robbins & Monro, 1951; Nemirovski et al., 2009; Bottou et al., 2018), which implicitly makes use of an *infinite* compute budget. That said, there exists a wide body of work (Zinkevich, 2003; Kingma & Ba, 2015; Reddi et al., 2018; Luo et al., 2019) that provide performance bounds which depend on the iteration number, which apply even in the non-asymptotic regime. Our work differs in its exploration of maximizing performance for a fixed number of iterations. Importantly, the globally optimal solution may not even be achievable in our budgeted setting.

Given a limited budget, one obvious strategy might be data subsampling (Bachem et al., 2017; Sener & Savarese, 2018). However, we discover that a much more effective, simpler, and under-explored strategy is adopting budget-aware learning rate schedules — if we know that we are limited to a single epoch, one should tune the learning schedule accordingly. Such budget-aware schedules have been proposed in previous work (Feyzmahdavian et al., 2016; Lian et al., 2017), but often for a fixed learning rate that depends on dataset statistics. In this paper, we specifically point out *linearly* decaying the learning rate to 0 at the end of the budget, may be more robust than more complicated strategies suggested in prior work. Though we are motivated by budget-aware training, we find that a linear schedule is quite competitive for general learning settings as well. We verify our findings with state-of-the-art models on ImageNet (image classification), Kinetics (video classification), MS COCO (object detection and instance segmentation), and Cityscapes (semantic segmentation).

We conduct several diagnostic experiments that analyze learning rate decays under the budgeted setting. We first observe a statistical correlation between the learning rate and the *full* gradient magnitude (over the entire dataset). Decreasing the learning rate empirically results in a decrease in the full gradient magnitude. Eventually, as the former goes to zero, the latter vanishes as well, suggesting that the optimization has reached a critical point, if not a local minimum[1]. We call this phenomenon *budgeted convergence* and we find it generalizes across budgets. On one hand, it implies that one should decay the learning rate to zero at the end of the training, even given a small budget. On the other hand, it implies one should not aggressively decay the learning rate early in the optimization (such as the case with an exponential schedule) since this may slow down later progress. Finally, we show that linear budget-aware schedules outperform recently-proposed fast-converging methods that make use of adaptive learning rates and restarts.

Our main contributions are as follows:

---

[1]Whether such a solution is exactly a local minimum or not is debatable (see Sec 2).

- We introduce a formal setting for budgeted training based on training iterations and provide an alternative perspective for existing learning rate schedules.
- We discover that budget-aware schedules are handy solutions to budgeted training. Specifically, our proposed linear schedule is more simple, robust, and effective than prior approaches, for both budgeted and general training.
- We provide an empirical justification of the effectiveness of learning rate decay based on the correlation between the learning rate and the full gradient norm.

## 2 RELATED WORK

**Learning rates.** Stochastic gradient descent dates back to Robbins & Monro (1951). The core is its update step: $w_t = w_{t-1} - \alpha_t g_t$, where $t$ (from 1 to $T$) is the iteration, $w$ are the parameters to be learned, $g$ is the gradient estimator for the objective function[2] $F$, and $\alpha_t$ is the *learning rate*, also known as *step size*. Given base learning rate $\alpha_0$, we can define the ratio $\beta_t = \alpha_t / \alpha_0$. Then the set of $\{\beta_t\}_{t=1}^T$ is called the *learning rate schedule*, which specifies how the learning rate should vary over the course of training. *Our definition differs slighter from prior art as it separates the base learning rate and learning rate schedule*. Learning rates are well studied for (strongly) convex cost surfaces and we include a brief review in Appendix H.

**Learning rate schedule for deep learning.** In deep learning, there is no consensus on the exact role of the learning rate. Most theoretical analysis makes the assumption of a small and constant learning rate (Du et al., 2018a;b; Hardt et al., 2016). For variable rates, one hypothesis is that large rates help move the optimization over large energy barriers while small rates help converge to a local minimum (Loshchilov & Hutter, 2017; Huang et al., 2017a; Kleinberg et al., 2018). Such hypothesis is questioned by recent analysis on mode connectivity, which has revealed that there does exist a descent path between solutions that were previously thought to be isolated local minima (Garipov et al., 2018; Dräxler et al., 2018; Gotmare et al., 2019). Despite a lack of theoretical explanation, the community has adopted a variety of heuristic schedules for practical purposes, two of which are particularly common:

- **step decay**: drop the learning rate by a multiplicative factor $\gamma$ after every $d$ epochs. The default for $\gamma$ is 0.1, but $d$ varies significantly across tasks.
- **exponential**: $\beta_t = \gamma^t$. There is no default parameter for $\gamma$ and it requires manual tuning.

State-of-the-art codebases for standard vision benchmarks tend to employ step decay (Xie & Tu, 2015; Huang et al., 2017b; He et al., 2017; Carreira & Zisserman, 2017; Wang et al., 2018; Yin et al., 2019; Ma et al., 2019), whereas exponential decay has been successfully used to train Inception networks (Szegedy et al., 2015; 2016; 2017). In spite of their prevalence, these heuristics have not been well studied. Recent work proposes several new schedules (Loshchilov & Hutter, 2017; Smith, 2017; Hsueh et al., 2019), but much of this past work limits their evaluation to CIFAR and ImageNet. For example, SGDR (Loshchilov & Hutter, 2017) advocates for learning-rate restarts based on the results on CIFAR, however, we find the unexplained form of cosine decay in SGDR is more effective than the restart technique. Notably, Mishkin et al. (2017) demonstrate the effectiveness of linear rate decay with CaffeNet on downsized ImageNet. In our work, we rigorously evaluate on 5 standard vision benchmarks with state-of-the-art networks and under various budgets. Gotmare et al. (2019) also analyze learning rate restarts and in addition, the warm-up technique, but do not analyze the specific form of learning rate decay.

**Adaptive learning rates.** Adaptive learning rate methods (Tieleman & Hinton, 2012; Kingma & Ba, 2015; Reddi et al., 2018; Luo et al., 2019) adjust the learning rate according to the local statistics of the cost surface. Despite having better theoretical bounds under certain conditions, they do not generalize as well as momentum SGD for benchmark tasks that are much larger than CIFAR (Wilson et al., 2017). We offer new insights by evaluating them under the budgeted setting. We show fast descent can be trivially achieved through budget-aware schedules and aggressive early descent is not desirable for achieving good performance in the end.

---

[2]Note that $g$ can be based on a single example, a mini-batch, the full training set, or the true data distribution. In most practical settings, momentum SGD is used, but we omit the momentum here for simplicity.

## 3 LEARNING RATES AND BUDGETS

### 3.1 BUDGET-AWARE SCHEDULES

Learning rate schedules are often defined assuming unlimited resources. As we argue, resource constraints are an undeniable practical aspect of learning. One simple approach for modifying an existing learning rate schedule to a budgeted setting is early-stopping. Fig 1 shows that one can dramatically improve results of early stopping by more than 60% by *tuning* the learning rate for the appropriate budget. To do so, we simply reparameterize the learning rate sequence with a quantity not only dependent on the absolute iteration $t$, but also the training budget $T$:

**Definition (Budget-Aware Schedule).** Let $T$ be the training budget, $t$ be the current step, then a training progress $p$ is $t/T$. A *budget-aware learning rate schedule* is

$$\beta_p : p \mapsto f(p), \tag{1}$$

where $f(p)$ is the ratio of learning rate at step $t$ to the base learning rate $\alpha_0$.

At first glance, it might be counter-intuitive for a schedule to *not* depend on $T$. For example, for a task that is usually trained with 200 epochs, training 2 epochs will end up at a solution very distant from the global optimal no matter the schedule. In such cases, conventional wisdom from convex optimization suggests that one should employ a large learning rate (constant schedule) that efficiently descends towards the global optimal. However, in the non-convex case, we observe empirically that a better strategy is to systematically decay the learning rate in proportion to the total iteration budget.

**Budge-Aware Conversion (BAC).** Given a particular rate schedule $\beta_t = f(t)$, one simple method for making it budget-aware is to rescale it, *i.e.*, $\beta_p = f(pT_0)$, where $T_0$ is the budget used for the original schedule. For instance, a step decay for 90 epochs with two drops at epoch 30 and epoch 60 will convert to a schedule that drops at 1/3 and 2/3 training progress. Analogously, an exponential schedule $0.99^t$ for 200 epochs will be converted into $(0.99^{200})^p$.

It is worth noting that such an adaptation strategy already exists in well-known codebases (He et al., 2017) for training with limited schedules. Our experiments confirm the effectiveness of BAC as a general strategy for converting many standard schedules to be budget-aware (Tab 1). *For our remaining experiments, we regard BAC as a known technique and apply it to our baselines by default.*

| Budget | 1% | 5% | 10% | 25% | 50% | 100% |
|---|---|---|---|---|---|---|
| exp .99 | .5848 | .8030 | .8352 | .8888 | .9072 | .9320 |
| BAC | **.6086** | **.8560** | **.8996** | **.9228** | **.9272** | N/A |
| step-d1 | .5710 | .8058 | .8422 | .8702 | .8746 | .9434 |
| BAC | **.5880** | **.8662** | **.9066** | **.9312** | **.9392** | N/A |

Table 1: Effectiveness of budget-aware conversion (BAC) on CIFAR-10 for image classification with ResNet-18 (He et al., 2016). The numbers are classification accuracy on the validation set. The 100% budget refers to training for 200 epochs. "step-d1" denotes step decay dropping once at training progress 50%. Please refer to Sec 4.1 for the complete setup.

**Recent schedules:** Interestingly, several recent learning rate schedules are implicitly defined as a function of progress $p = \frac{t}{T}$, and so are budget-aware by our definition:

- **poly** (Jia et al., 2014): $\beta_p = (1-p)^\gamma$. No parameter other than $\gamma = 0.9$ is used in published work.
- **cosine** (Loshchilov & Hutter, 2017): $\beta_p = \eta + \frac{1}{2}(1 - \eta)(1 + \cos(\pi p))$. $\eta$ specify a lower bound for the learning rate, which defaults to zero.
- **htd** (Hsueh et al., 2019): $\beta_p = \eta + \frac{1}{2}(1 - \eta)(1 - \tanh(L + (U - L)p))$. Here $\eta$ has the same representation as in cosine. It is reported that $L = -6$ and $U = 3$ performs the best.

The poly schedule is a feature in Caffe (Jia et al., 2014) and adopted by the semantic segmentation community (Chen et al., 2018; Zhao et al., 2017). The cosine schedule is a byproduct in work that promotes learning rate restarts (Loshchilov & Hutter, 2017). The htd schedule is recently proposed (Hsueh et al., 2019), which however, contains only limited empirical evaluation. None of these analyze their budget-aware property or provides intuition for such forms of decay. These schedules were

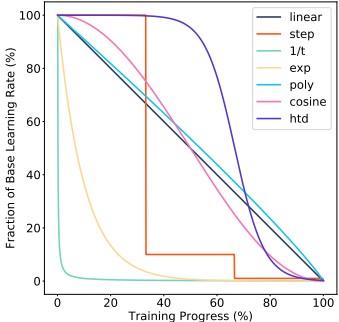 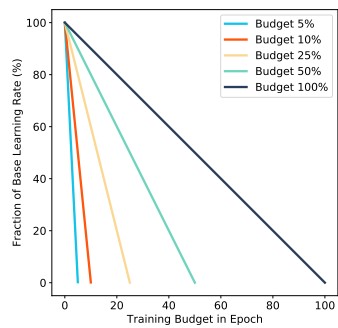

Figure 2: We normalize various learning rate schedules by training progress (left). Our solution to budgeted training is simple and universal — we decrease the learning rate linearly across the entire given budget (right).

| Budget | 1% | 5% | 10% | 25% | 50% | 100% |
|--------|-----|-----|------|------|------|-------|
| const | .5748 ± .0337 | .7989 ± .0093 | .8350 ± .0122 | .8658 ± .0007 | .8723 ± .0044 | .8767 ± .0066 |
| exp .95 | .4834 ± .0125 | .7575 ± .0053 | .8567 ± .0027 | .9147 ± .0030 | .9295 ± .0006 | .9468 ± .0021 |
| exp .97 | .5467 ± .0202 | .8348 ± .0016 | .8936 ± .0030 | .9294 ± .0024 | .9413 ± .0015 | .9551 ± .0004 |
| exp .99 | .6069 ± .0219 | .8557 ± .0037 | .9013 ± .0036 | .9227 ± .0033 | .9268 ± .0026 | .9310 ± .0023 |
| step-d1 | .5853 ± .0134 | .8643 ± .0027 | .9063 ± .0023 | .9307 ± .0020 | .9423 ± .0027 | .9426 ± .0031 |
| step-d2 | .5487 ± .0156 | .8342 ± .0052 | .9043 ± .0034 | .9319 ± .0037 | .9461 ± .0019 | .9529 ± .0009 |
| step-d3 | .4879 ± .0036 | .7929 ± .0061 | .8864 ± .0027 | .9259 ± .0006 | .9437 ± .0001 | .9527 ± .0019 |
| htd | .6450 ± .0070 | .8899 ± .0043 | .9219 ± .0014 | .9449 ± .0031 | .9520 ± .0023 | .9554 ± .0013 |
| cosine | .6343 ± .0080 | .8851 ± .0024 | .9223 ± .0024 | .9432 ± .0024 | .9520 ± .0026 | .9552 ± .0021 |
| poly | .6595 ± .0086 | .8905 ± .0017 | .9247 ± .0008 | .9421 ± .0019 | .9494 ± .0034 | .9540 ± .0012 |
| linear | .6617 ± .0079 | .8915 ± .0011 | .9217 ± .0028 | .9412 ± .0018 | .9537 ± .0020 | .9563 ± .0009 |

Table 2: Comparison of learning rate schedules on CIFAR-10. The 1st, 2nd and the 3rd place under each budget are color coded. The number here is the classification accuracy and each one is the average of 3 independent runs. "step-d$x$" denotes decay $x$ times at even intervals with $\gamma = 0.1$. For "exp" and "step" schedules, BAC (Sec 3.1) is applied in place of early stopping. We can see linear schedule surpasses other schedules under almost all budgets.

treated as "yet another schedule". However, our definition of budget-aware makes these schedules stand out as a general family.

## 3.2 LINEAR SCHEDULE

Inspired by existing budget-aware schedules, we borrow an even simpler schedule from the simulated annealing literature (Kirkpatrick et al., 1983; McAllester et al., 1997; Nourani & Andresen, 1998)[3]:

$$\textbf{linear} : \beta_p = 1 - p. \tag{2}$$

In Fig 2, we compare linear schedule with various existing schedules under the budget-aware setting. Note that this linear schedule is completely parameter-free. This property is particularly desirable in budgeted training, where little budget exists for tuning such a parameter. The excellent generalization of linear schedule across budgets (shown in the next section) might imply that the cost surface of deep learning is to some degree self-similar. Note that a linear schedule, together with

---

[3]A link between SGD and simulated annealing has been recognized decades ago, where learning rate plays the role of temperature control (Bottou, 1991). Therefore, cooling schedules in simulated annealing can be transferred into learning rate schedules for SGD.

other recent budget-aware schedules, produces a constant learning rate in the asymptotic limit *i.e.*, $\lim_{T \to \infty}(1 - t/T) = 1$. Consequently, such practically high-performing schedules tend to be ignored in theoretical convergence analysis (Robbins & Monro, 1951; Bottou et al., 2018).

## 4 EXPERIMENTS

In this section, we first compare linear schedule against other existing schedules on the small CIFAR-10 dataset and then on a broad suite of vision benchmarks. The CIFAR-10 experiment is designed to extensively evaluate each learning schedule while the vision benchmarks are used to verify the observation on CIFAR-10. We provide important implementation settings in the main text while leaving the rest of the details to Appendix K. In addition, we provide in Appendix A the evaluation with a large number of random architectures in the setting of neural architecture search.

### 4.1 CIFAR

CIFAR-10 (Krizhevsky & Hinton, 2009) is a dataset that contains 70,000 tiny images ($32 \times 32$). Given its small size, it is widely used for validating novel architectures. We follow the standard setup for dataset split (Huang et al., 2017b), which is randomly holding out 5,000 from the 50,000 training images to form the validation set. For each budget, we report the best validation accuracy among epochs up till the end of the budget. We use ResNet-18 (He et al., 2016) as the backbone architecture and utilize SGD with base learning rate 0.1, momentum 0.9, weight decay 0.0005 and a batch size 128.

We study learning schedules in several groups: (a) constant (equivalent to not using any schedule). (b) & (c) exponential and step decay, both of which are commonly adopted schedules. (d) htd (Hsueh et al., 2019), a quite recent addition and not yet adopted in practice . We take the parameters with the best-reported performance $(-6, 3)$. Note that this schedule decays much slower initially than the linear schedule (Fig 2). (e) the smooth-decaying schedules (small curvature), which consists of cosine (Loshchilov & Hutter, 2017), poly (Jia et al., 2014) and the linear schedule.

As shown in Tab 2, the group of schedules that are budget-aware by our definition, outperform other schedules under all budgets. The linear schedule in particular, performs best most of the time including the typical full budget case. Noticeably, when exponential schedule is well-tuned for this task ($\gamma = 0.97$), it fails to generalize across budgets. In comparison, the budget-aware group does not require tuning but generalizes much better.

Within the budget-aware schedules, cosine, poly and linear achieve very similar results. This is expected due to the fact that their numerical similarity at each step (Fig 2). These results might indicate that *the key for a robust budgeted-schedule is to decay smoothly to zero.* Based on these observations and results, we suggest linear schedule should be the "go-to" budget-aware schedule.

### 4.2 VISION BENCHMARKS

In the previous section, we showed that linear schedule achieves excellent performance on CIFAR-10, in a relatively toy setting. In this section, we study the comparison and its generalization to practical large scale datasets with various state-of-the-art architectures. In particular, we set up experiments to validate the performance of linear schedule across tasks and budgets.

Ideally, one would like to see the performance of all schedules in Fig 2 on vision benchmarks. Due to resource constraints, we include only the off-the-shelf step decay and the linear schedule. Note our CIFAR-10 experiment suggests that using cosine and poly will achieve similar performance as linear, which are already budget-aware schedules given our definition, so we focus on linear schedule in this section. More evaluation between cosine, poly and linear can be found in Appendix A & D.

We consider the following suite of benchmarks spanning many flagship vision challenges:

**Image classification on ImageNet.** ImageNet (Russakovsky et al., 2015) is a widely adopted standard for image classification task. We use ResNet-18 (He et al., 2016) and report the top-1 accuracy on the validation set with the best epoch. We follow the step decay schedule used in (Huang et al., 2017b; PyTorch, 2019), which drops twice at uniform interval ($\gamma = 0.1$ at $p \in \{\frac{1}{3}, \frac{2}{3}\}$). We set the full budget to 100 epochs (10 epochs longer than typical) for easier computation of the budget.

| Budget | 1% | 5% | 10% | 25% | 50% | 100% |
|---|---|---|---|---|---|---|
| | | | Image classification on ImageNet with ResNet | | | |
| step | .2039 ± .0029 | .5194 ± .0048 | .5951 ± .0021 | .6558 ± .0018 | .6796 ± .0008 | **.6934** ± .0018 |
| linear | **.3063** ± .0036 | **.5726** ± .0024 | **.6232** ± .0004 | **.6634** ± .0020 | **.6818** ± .0013 | .6933 ± .0012 |
| | | | Object detection on COCO with Mask-RCNN | | | |
| step | .0486 ± .0024 | .2003 ± .0008 | .2541 ± .0005 | .3149 ± .0015 | .3530 ± .0005 | .3767 ± .0009 |
| linear | **.0513** ± .0042 | **.2090** ± .0016 | **.2626** ± .0008 | **.3222** ± .0014 | **.3572** ± .0003 | **.3795** ± .0012 |
| | | | Instance segmentation on COCO with Mask-RCNN | | | |
| step | .0487 ± .0029 | .1925 ± .0004 | .2388 ± .0007 | .2907 ± .0003 | .3202 ± .0009 | .3395 ± .0009 |
| linear | **.0507** ± .0040 | **.1986** ± .0012 | **.2457** ± .0007 | **.2942** ± .0002 | **.3242** ± .0005 | **.3396** ± .0009 |
| | | | Semantic segmentation on Cityscapes with PSPNet | | | |
| step | .4941 ± .0011 | .6358 ± .0052 | .6800 ± .0010 | .7250 ± .0019 | .7423 ± .0094 | **.7651** ± .0032 |
| linear | **.5424** ± .0034 | **.6654** ± .0014 | **.7076** ± .0047 | **.7399** ± .0005 | **.7575** ± .0041 | .7633 ± .0008 |
| | | | Video classification on Kinetics with I3D | | | |
| step | .2941 ± .0028 | .4981 ± .0029 | .5674 ± .0013 | .6459 ± .0023 | .6870 ± .0025 | .7134 ± .0021 |
| linear | **.3286** ± .0042 | **.5297** ± .0014 | **.5967** ± .0030 | **.6634** ± .0020 | **.6995** ± .0011 | **.7223** ± .0031 |

Table 3: Robustness of linear schedule across budgets, tasks and architectures. Linear schedule significantly outperforms step decay given limited budgets. Note that the off-the-shelf decay for each dataset has different parameters optimized for the specific dataset. For all step decay schedules, BAC (Sec 3.1) is applied to boost their budgeted performance. To reduce stochastic noise, we report the average and the standard deviation of 3 independent runs. See Sec 4.2 for the metrics of each task (the higher the better for all tasks).

**Object detection and instance segmentation on MS COCO.** MS COCO (Lin et al., 2014) is a widely recognized benchmark for object detection and instance segmentation. We use the standard COCO AP (averaged over IoU thresholds) metric for evaluating bounding box output and instance mask output. The AP of the final model on the validation set is reported in our experiment. We use the challenge winner Mask R-CNN (He et al., 2017) with a ResNet-50 backbone and follow its setup. For training, we adopt the 1x schedule (90k iterations), and the off-the-shelf (He et al., 2017) step decay that drops 2 times with $\gamma = 0.1$ at $p \in \{\frac{2}{3}, \frac{8}{9}\}$.

**Semantic segmentation on Cityscapes.** Cityscapes (Cordts et al., 2016) is a dataset commonly used for evaluating semantic segmentation algorithms. It contains high quality pixel-level annotations of 5k images in urban scenarios. The default evaluation metric is the mIoU (averaged across class) of the output segmentation map. We use state-of-the-art model PSPNet (Zhao et al., 2017) with a ResNet-50 backbone and the full budget is 400 epochs as in standard set up. The mIoU of the best epoch is reported. Interestingly, unlike other tasks in this series, this model by default uses the poly schedule. For complete evaluation, we add step decay that is the same in our ImageNet experiment in Tab 3 and include the off-the-shelf poly schedule in Tab E.

**Video classification on Kinetics with I3D.** Kinetics (Kay et al., 2017) is a large-scale dataset of YouTube videos focusing on human actions. We use the 400-category version of the dataset and a variant of I3D (Carreira & Zisserman, 2017) with training and data processing code publicly available (Wang et al., 2018). The top-1 accuracy of the final model is used for evaluating the performance. We follow the 4-GPU 300k iteration schedule (Wang et al., 2018), which features a step decay that drops 2 times with $\gamma = 0.1$ at $p \in \{\frac{1}{2}, \frac{5}{6}\}$.

If we factor in the dimension of budgets, Tab 3 shows a clear advantage of linear schedule over step decay. For example, on ImageNet, linear achieves 51.5% improvement at 1% of the budget. Next, we consider the full budget setting, where we simply swap out the off-the-shelf schedule with linear schedule. We observe better (video classification) or comparable (other tasks) performance after the swap. This is surprising given the fact that linear schedule is parameter-free and thus not optimized for the particular task or network.

In summary, *the smoothly decaying linear schedule is a simple and effective strategy for budgeted training*. It significantly outperforms traditional step decay given limited budgets, while achieving comparable performance with the normal full budget setting.

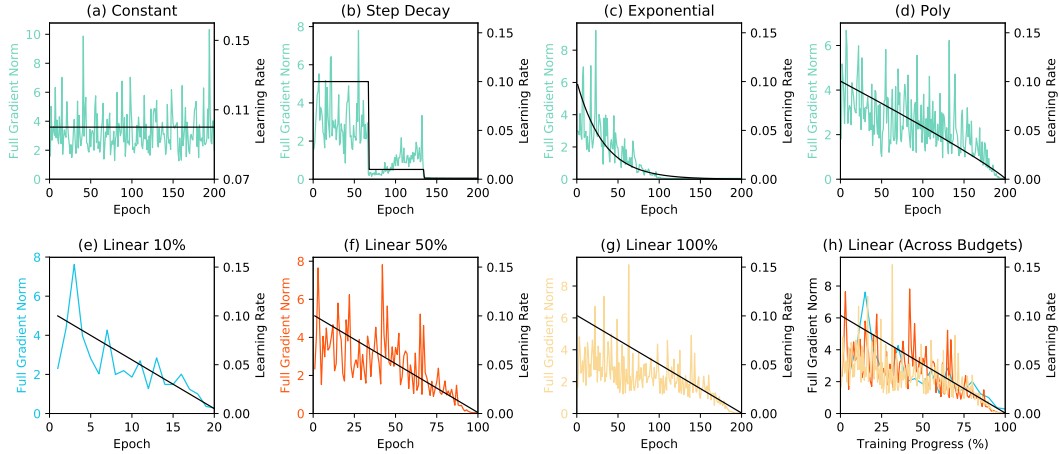

Figure 3: Budgeted convergence: full gradient norm $||g_t^*||$ vanishes over time (color curves) as learning rate $\alpha_t$ (black curves) decays. The first row shows that the dynamics of full gradient norm correlate with the corresponding learning rate schedule while the second row shows that such phenomena generalize across budgets for budget-aware schedules. Such generalization is most obvious in plot (h), which overlays the full gradient norm across different budgets. If a schedule does not decay to 0, the gradient norm does not vanish. For example, if we train a budget-unaware exponential schedule for 50 epochs (c), the full gradient norm at that time is around 1.5, suggesting this is a schedule with insufficient final decay of learning rate.

## 5 DISCUSSION

In this section, we summarize our empirical analysis with a *desiderata* of properties for effective budget-aware learning schedules. We highlight those are inconsistent with conventional wisdom and follow the experimental setup in Sec 4.1 unless otherwise stated.

**Desideratum: budgeted convergence.** Convergence of SGD under non-convex objectives is measured by $\lim_{t\to\infty} \mathbb{E}[||\nabla F||^2] = 0$ (Bottou et al., 2018). Intuitively, one should terminate the optimization when no further local improvement can be made. What is the natural counterpart for "convergence" within a budget? For a dataset of $N$ examples $\{(x_i, y_i)\}_{i=1}^N$, let us write the full gradient as $g_t^* = \frac{1}{N}\sum_{i=1}^N \nabla F(x_i, y_i)$. We empirically find that the dynamics of $||g_t^*||$ over time highly correlates with the learning rate $\alpha_t$ (Fig 3). As the learning rate vanishes for budget-aware schedules, so does the gradient magnitude. We call this "vanishing gradient" phenomenon *budgeted convergence*. This correlation suggests that decaying schedules to near-zero rates (and using BAC) may be more effective than early stopping. As a side note, budgeted convergence resonates with classic literature that argues that SGD behaves similar to simulated annealing (Bottou, 1991). Given that $\alpha_t$ and $||g_t^*||$ decrease, the overall update $||-\alpha_t g_t||$ also decreases[4]. In other words, large moves are more likely given large learning rates in the beginning, while small moves are more likely given small learning rates in the end. However, the exact mechanism by which the learning rate influences the gradient magnitude remains unclear.

**Desideratum: don't waste the budget.** Common machine learning practise often produces multiple checkpointed models during a training run, where a validation set is used to select the best one. Such additional optimization is wasteful in our budgeted setting. Tab 4 summarizes the progress point at which the best model tends to be found. Step decay produces an optimal model somewhat towards the end of the training, while linear and poly are almost always optimal at the precise end of the training. This is especially helpful for state-of-the-art models where evaluation can be expensive. For example, validation for Kinetics video classification takes several hours. Budget-aware schedules require validation on only the last few epochs, saving additional compute.

---

[4]Note that the momentum in SGD is used, but we assume vanilla SGD to simplify the discussion, without losing generality.

| Schedule | Best Progress | Schedule | Best Progress |
|----------|---------------|----------|---------------|
| const | $81.2\% \pm 16.1\%$ | step-d2 | $90.5\% \pm 9.0\%$ |
| linear | $98.6\% \pm 1.6\%$ | poly | $99.1\% \pm 1.3\%$ |

Table 4: Where does one expect to find the model with the highest validation accuracy within the training progress? Here we show the best checkpoint location measured in training progress $p$ and averaged for each schedule across budgets greater or equal than 10% and 3 different runs.

**Aggressive early descent**. Guided by asymptotic convergence analysis, faster descent of the objective might be an apparent desideratum of an optimizer. Many prior optimization methods explicitly call for faster decrease of the objective (Kingma & Ba, 2015; Clevert et al., 2016; Reddi et al., 2018). In contrast, we find that one should not employ aggressive early descent because large learning rates can prevent budgeted convergence. Consider AMSGrad (Reddi et al., 2018), an adaptive learning rate that addresses a convergence issue with the widely-used Adam optimizer (Kingma & Ba, 2015). Fig 4 shows that while AMSGrad does quickly descend over the training objective, it still underperforms budget-aware linear schedules over *any* given training budget. To examine why, we derive the equivalent rate $\widetilde{\beta}_t$ for AMSGrad (Appendix B) and show that it is dramatically larger than our defaults, suggesting the optimizer is too aggressive. We include more adaptive methods for evaluation in Appendix E.

**Warm restarts**. SGDR (Loshchilov & Hutter, 2017) explores periodic schedules, in which each period is a cosine scaling. The schedule is intended to escape local minima, but its effectiveness has been questioned (Gotmare et al., 2019). Fig 5 shows that SDGR has faster descent but is inferior to budget-aware schedules for *any* budget (similar to the adaptive optimizers above). Additional comparisons can be found in Appendix F. Whether there exists a method that achieves promising anytime performance and budgeted performance at the same time remains an open question.

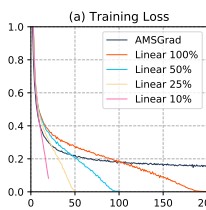 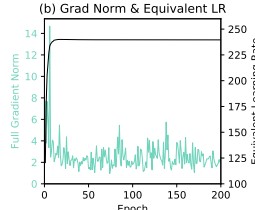 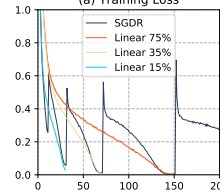 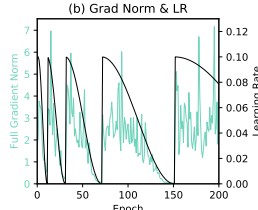

Figure 4: Comparing AMSGrad (Reddi et al., 2018) with linear schedule. (a) while AMSGrad makes fast initial descent of the training loss, it is surpassed at each given budget by the linear schedule. (b) budgeted convergence is not observed for AMSGrad — the full gradient norm $||g_t^*||$ does not vanish (color curves). Comparing to a momentum SGD, AMSGrad recommends magnitudes larger learning rate $\widetilde{\beta}_t$ (black curve).

Figure 5: Comparing SGDR (Loshchilov & Hutter, 2017) with linear schedules. (a) SGDR makes slightly faster initial descent of the training loss, but is surpassed at each given budget by the linear schedule. (b) for SGDR, the correlation between full gradient norm $||g_t^*||$ and learning rate $\alpha_t$ is also observed. Warm restart does not help to achieve better budgeted performance.

## 6 CONCLUSION

This paper introduces a formal setting for budgeted training. Under this setup, we observe that a simple linear schedule, or any other smooth-decaying schedules can achieve much better performance. Moreover, the linear schedule even offers comparable performance on existing visual recognition tasks for the typical full budget case. In addition, we analyze the intriguing properties of learning rate schedules under budgeted training. We find that the learning rate schedule controls the gradient magnitude regardless of training stage. This further suggests that SGD behaves like simulated annealing and the purpose of a learning rate schedule is to control the stage of optimization.

**Acknowledgements:** We thank Xiaofang Wang, Simon S. Du, Leonid Keselman, Chen-Hsuan Lin and David McAllester for insightful discussions and comments.

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

# A    BUDGETED TRAINING FOR NEURAL ARCHITECTURE SEARCH

## A.1    RANK PREDICTION

In the main text, we list neural architecture search as an application of budgeted training. Due to resource constraint, these methods usually train models with a small budget (10-25 epochs) to evaluate their relative performance (Cao et al., 2019; Cai et al., 2018; Real et al., 2019). Under this setting, *the goal is to rank the performance of different architectures* instead of obtaining the best possible accuracy as in the regular case of budgeted training. Then one could ask the question that whether budgeted training techniques help in better predicting the relative rank. Unfortunately, budgeted training has not been studied or discussed in the neural architecture search literature, it is unknown how well models only trained with 10 epochs can tell the relative performance of the same ones that are trained with 200 epochs. Here we conduct a controlled experiment and show that proper adjustment of learning schedule, specifically the linear schedule, indeed improves the accuracy of rank prediction.

We adapt the code in (Cao et al., 2019) to generate 100 random architectures, which are obtained by random modifications (adding skip connection, removing layer, changing filter numbers) on top of ResNet-18 (He et al., 2017). First, we train these architectures on CIFAR-10 given full budget (200 epochs), following the setting described in Sec 4.1. This produces a relative rank between all pairs of random architectures based on the validation accuracy and this rank is considered as the target to predict given limited budget. Next, every random architecture is trained with various learning schedules under various small budgets. For each schedule and each budget, this generates a complete rank. We treat this rank as the prediction and compare it with the target full-budget rank. The metric we adopt is Kendall's rank correlation coefficient ($\tau$), a standard statistics metric for measuring rank similarity. It is based on counting the inversion pairs in the two ranks and $(\tau + 1)/2$ is approximately the probability of estimating the rank correctly for a pair.

We consider the following schedules: (1) constant, it might be possible that no learning rate schedule is required if only the relative performance is considered. (2) step decay ($\gamma = 0.1$, decay at $p \in \{\frac{1}{3}, \frac{2}{3}\}$), a schedule commonly used both in regular training and neural architecture search (Zoph & Le, 2017; Pham et al., 2018). (3) cosine, a schedule often used in neural architecture search (Cai et al., 2018; Real et al., 2019). (4) linear, our proposed schedule. The results of their rank prediction capability can be seen in Tab A.

The results suggest that with more budget, we can better estimate the full-budget rank between architectures. And even if only relative performance is considered, learning rate decay should be applied. Specifically, smooth-decaying schedule, such as linear or cosine, are preferred over step decay.

We list some additional details about the experiment. To reduce stochastic noise, each configuration under both the small and full budget is repeated 3 times and the median accuracy is taken. The full-budget model is trained with linear schedule, similar results are expected with other schedules as evidenced by the CIFAR-10 results in the main text (Tab 2). Among the 100 random architectures, 21 cannot be trained, the rest of 79 models have validation accuracy spanning from 0.37 to 0.94, with the distribution mass centered at 0.91. Such skewed and widespread distribution is the typical case in neural architecture search. We remove the 21 models that cannot be trained for our experiments. We take the epoch with the best validation accuracy for each configuration, so the drawback of constant or step decay not having the best model at the very end does not affect this experiment (see Sec 5).

| Epoch (Budget) | 1 (0.5%) | 2 (1%) | 10 (5%) | 20 (10%) |
|---|---|---|---|---|
| const | 0.3451 | 0.4595 | 0.6720 | 0.6926 |
| step-d2 | 0.2746 | 0.3847 | 0.6651 | 0.7279 |
| cosine | 0.3211 | **0.4847** | 0.7023 | **0.7563** |
| linear | **0.3409** | 0.4348 | **0.7398** | 0.7351 |

Table A: Small-budget and full-budget model rank correlation measured in Kendall's tau. Smooth-decaying schedules like linear and cosine can more accurately predict the true rank of different architectures given limited budget.

| Epoch (Budget) | 1 (0.5%) | 2 (1%) | 10 (5%) | 20 (10%) |
|---|---|---|---|---|
| const | 0.3892 | 0.4699 | 0.6689 | 0.7061 |
| step-d2 | 0.4014 | 0.4780 | 0.6980 | 0.7754 |
| cosine | 0.4616 | 0.5498 | 0.7530 | 0.8029 |
| linear | **0.4759** | **0.5745** | **0.7652** | **0.8192** |

Table B: Small-budget validation accuracy averaged across random architectures. Linear schedule is the most robust under small budgets.

| Epoch (Budget) | 1 (0.5%) | 2 (1%) | 10 (5%) | 20 (10%) |
|---|---|---|---|---|
| const | 0.4419 | 0.5343 | 0.7550 | 0.8015 |
| step-d2 | 0.4590 | 0.5455 | 0.7894 | 0.8848 |
| cosine | 0.5326 | 0.6265 | 0.8615 | 0.9087 |
| linear | **0.5431** | **0.6626** | **0.8644** | **0.9305** |

Table C: Tab B normalized by the full-budget accuracy and then averaged across architectures. Linear schedule achieves solutions closer to their full-budget performance than the rest of schedules under small budgets.

## A.2 BUDGETED PERFORMANCE ACROSS ARCHITECTURES

To reinforce our claim that linear schedule generalizes across different settings, we compare budgeted performance of various schedules on random architectures generated in the previous section. We present two versions of the results. The first is to directly average the validation accuracy of different architecture with each schedule and under each budget (Tab B). The second is to normalize by dividing the budgeted accuracy by the full-budget accuracy of the same architecture and then average across different architectures (Tab C). The second version assumes all architectures enjoy equal weighting. Under both cases, linear schedule is the most robust schedule across architectures under various budgets.

## B EQUIVALENT LEARNING RATE FOR AMSGRAD

In Sec 5, we use equivalent learning rate to compare AMSGrad (Reddi et al., 2018) with momentum SGD. Here we present the derivation for the equivalent learning rate $\widetilde{\beta}_t$.

Let $\eta_1$, $\eta_2$ and $\epsilon$ be hyper-parameters, then the momentum SGD update rule is:

$$m_t = \eta_1 m_{t-1} + (1 - \eta_1)g_t, \tag{3}$$

$$w_t = w_{t-1} - \alpha_0^{(1)}\beta_t m_t, \tag{4}$$

while the AMSGrad update rule is:

$$m_t = \eta_1 m_{t-1} + (1 - \eta_1)g_t, \tag{5}$$

$$v_t = \eta_2 v_{t-1} + (1 - \eta_2)g_t^2, \tag{6}$$

$$\hat{m}_t = \frac{m_t}{1 - \eta_1^t}, \tag{7}$$

$$\hat{v}_t = \frac{v_t}{1 - \eta_2^t}, \tag{8}$$

$$\hat{v}_t^{\max} = \max(\hat{v}_t^{\max}, \hat{v}_t) \tag{9}$$

$$w_t = w_{t-1} - \alpha_0^{(2)}\frac{\hat{m}_t}{\sqrt{\hat{v}_t^{\max}} + \epsilon}. \tag{10}$$

Comparing equation 4 with 10, we obtain the equivalent learning rate:

$$\widetilde{\beta}_t = \frac{\alpha_0^{(2)}}{\alpha_0^{(1)}}\frac{1}{(1 - \eta_1^t)(\sqrt{\hat{v}_t^{\max}} + \epsilon)}, \tag{11}$$

| Budget | 1% | 5% | 10% | 25% | 50% | 100% |
|--------|------|------|------|------|------|------|
| Subset | .3834 | .6446 | .7848 | .8586 | .9234 | N/A |
| Full | **.5544** | **.8328** | **.9042** | **.9338** | **.9464** | **.9534** |

Table D: Comparison with offline data subsampling. "Subset" meets the budget constraint by randomly subsample the dataset prior to training, while "full" uses all the data, but restricting the number of iterations. Note that budget-aware schedule is used for "full".

Note that the above equation holds per each weight. For Fig 4a, we take the median across all dimensions as a scalar summary since it is a skewed distribution. The mean appears to be even larger and shares the same trend as the median. In our experiments, we use the default hyper-parameters (which also turn out to have the best validation accuracy): $\alpha_0^{(1)} = 0.1$, $\alpha_0^{(2)} = 0.001$, $\eta_1 = 0.9$, $\eta_2 = 0.99$ and $\epsilon = 10^{-8}$.

## C  DATA SUBSAMPLING

Data subsampling is a straight-forward strategy for budgeted training and can be realized in several different ways. In our work, we limit the number of iterations to meet the budget constraint and this effectively limits the number of data points seen during the training process. An alternative is to construct a subsampled dataset offline, but keep the same number of training iterations. Such construction can be done by random sampling, which might be the most effective strategy for *i.i.d* (independent and identically distributed) dataset. We show in Tab D that even our baseline budge-aware step decay, together with a limitation on the iterations, can significantly outperform this offline strategy. For the subset setting, we use the off-the-shelf step decay (step-d2) while for the full set setting, we use the same step decay but with BAC applied (Sec 3.1). For detailed setup, we follow Sec 4.1, of the main text.

Of course, more complicated subset construction methods exist, such as *core-set* construction (Bachem et al., 2017). However, such methods usually requires a feature summary of each data point and the computation of pairwise distance, making such methods unsuitable for extremely large dataset. In addition, note that our subsampling experiment is conducted on CIFAR-10, a well-constructed and balanced dataset, making smarter subsampling methods less advantageous. Consequently, the result in Tab D can as well provides a reasonable estimate for other complicated subsampling methods.

## D  ADDITIONAL EXPERIMENTS ON CITYSCAPES (SEMANTIC SEGMENTATION)

In the main text, we compare linear schedule against step decay for various tasks. However, the off-the-shelf schedule for PSPNet (Zhao et al., 2017) is poly instead of step decay. Therefore, we include the evaluation of poly schedule on Cityscapes (Cordts et al., 2016) in Tab E. Given the similarity of poly and linear (Fig 2), and the opposite results on CIFAR-10 and Cityscapes, it is inconclusive that one is strictly better than the other within the smooth-decaying family. However, these smooth-decaying methods both outperform step decay given limited budgets.

| Budget | 1% | 5% | 10% | 25% | 50% | 100% |
|--------|------|------|------|------|------|------|
| poly | **.5476** ± .0023 | **.6755** ± .0012 | **.7093** ± .0058 | **.7416** ± .0028 | .7562 ± .0045 | .7593 ± .0043 |
| linear | .5424 ± .0034 | .6654 ± .0014 | .7076 ± .0047 | .7399 ± .0005 | **.7575** ± .0041 | **.7633** ± .0008 |

Table E: Comparison with off-the-shelf poly schedule on Cityscapes Cordts et al. (2016) using PSP-Net Zhao et al. (2017). Poly and linear are similar smooth-decaying schedules (Fig 2) and thus have similar performance. The exact rank differs from task to task.

# E    ADDITIONAL COMPARISON WITH ADAPTIVE LEARNING RATES

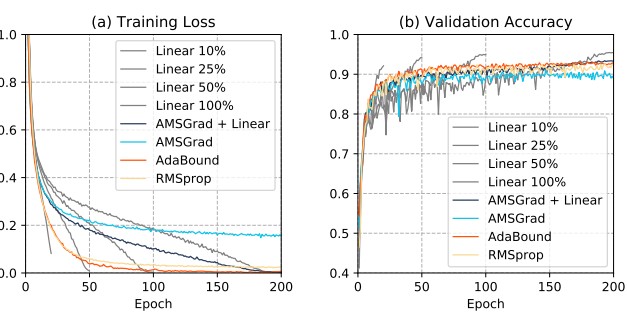

Figure A: Comparison between budget-aware linear schedule and adaptive learning rate methods on CIFAR-10. We see while adaptive learning rate methods appear to descent faster than full budget linear schedule, at each given budget, they are surpassed by the corresponding linear schedule.

In the main text we compare linear schedule with AMSGrad (Reddi et al., 2018) (the improved version over Adam (Kingma & Ba, 2015)), we further include the classical method RMSprop (Tieleman & Hinton, 2012) and the more recent AdaBound (Luo et al., 2019). We tune these adaptive methods for CIFAR-10 and summarize the results in Fig A. We observe the similar conclusion that budget-aware linear schedule outperforms adaptive methods for all given budgets.

Like SGD, those adaptive learning rate methods also takes input a parameter of base learning rate, which can also be annealed using an existing schedule. Although it is unclear why one needs to anneal an adaptive methods, we find that it in facts boosts the performance ("AMSGrad + Linear" in Fig A).

# F    ADDITIONAL COMPARISON WITH SGDR

This section provides additional evaluation to show that learning rate restart produces worse results than our proposed budgeted training techniques under budgeted setting. In (Loshchilov & Hutter, 2017), both a new form of decay (cosine) and the technique of learning rate restart are proposed. To avoid confusion, we use "cosine schedule", or just "cosine", to refer to the form of decay and SGDR to a schedule of periodical cosine decays. The comparison with cosine schedule is already included in the main text. Here we focus on evaluating the periodical schedule. SGDR requires two parameters to specify the periods: $T_0$, the length of the first period; $T_{\mathrm{mult}}$, where $i$-th period has length $T_i = T_0 T_{\mathrm{mult}}^{i-1}$. In Fig B, we plot the off-the-shelf SGDR schedule with $T_0 = 10$ (epoch), $T_{\mathrm{mult}} = 2$. The validation accuracy plot (on the right) shows that it might end at a very poor solution (0.8460) since it is not budget-aware. Therefore, we consider two settings to compare

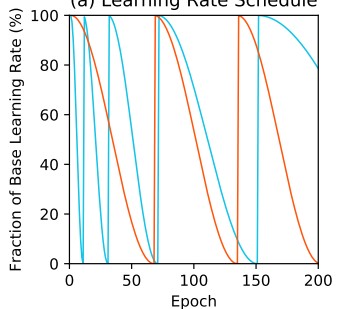 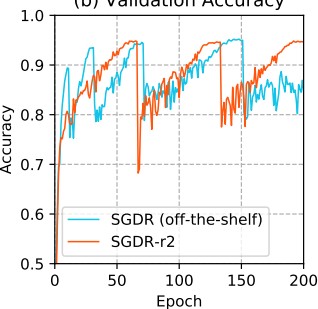

Figure B: One issue with off-the-shelf SGDR ($T_0 = 10$, $T_{\mathrm{mult}} = 2$) is that it is not budget-aware and might end at a poor solution. We convert it to a budget aware schedule by setting it to restart $n$ times at even intervals across the budget and $n = 2$ is shown here (SGDR-r2).

| Epoch | 30 | 50 | 150 |
|---|---|---|---|
| SGDR | .9320 | .9458 | .9510 |
| linear | **.9350** | **.9506** | **.9532** |

Table F: Comparison with off-the-shelf SGDR at the end of each period after the first restart.

| Budget | 1% | 5% | 10% | 25% | 50% | 100% |
|---|---|---|---|---|---|---|
| SGDR-r1 | .5002 | .7908 | .8794 | .9250 | .9380 | .9488 |
| SGDR-r2 | .4710 | .7888 | .8738 | .9216 | .9412 | .9502 |
| linear | **.6654** | **.8920** | **.9218** | **.9412** | **.9546** | **.9562** |

Table G: Comparison with SGDR under budget-aware setting. "SGDR-r1" refers to restarting learning rate once at midpoint of the training progress, and "SGDR-r2" refers to restarting twice at even interval.

linear schedule with SGDR. The first is to compare only at the end of each period of SGDR, where budgeted convergence is observed. The second is to convert SGDR into a budget-aware schedule by setting the schedule to restart $n$ times at even intervals across the budget. The results under the first and second setting is shown in Tab F and Tab G respectively. Under both budget-aware and budget-unaware setting, linear schedule outperforms SGDR. For detailed setup, we follow Sec 4.1, of the main text and take the median of 3 runs.

## G ADDITIONAL ILLUSTRATIONS

In Sec 5, we refer to validation accuracy curve for training on CIFAR-10, which we provide here in Fig C.

## H LEARNING RATES IN CONVEX OPTIMIZATION

For convex cost surfaces, constant learning rates are guaranteed to converge when less or equal than $1/L$, where $L$ is the Lipschitz constant for the gradient of the cost function $\nabla F$ (Bottou et al., 2018). Another well-known result ensures convergence for sequences that decay neither too fast nor too slow (Robbins & Monro, 1951): $\sum_{t=1}^{\infty} \alpha_t = \infty, \sum_{t=1}^{\infty} \alpha_t^2 < \infty$. One common such instance in convex optimization is $\alpha_t = \alpha_0/t$. For non-convex problems, similar results hold for convergence to a local minimum (Bottou et al., 2018). Unfortunately, there does not exist a theory for learning rate schedules in the context of general non-convex optimization.

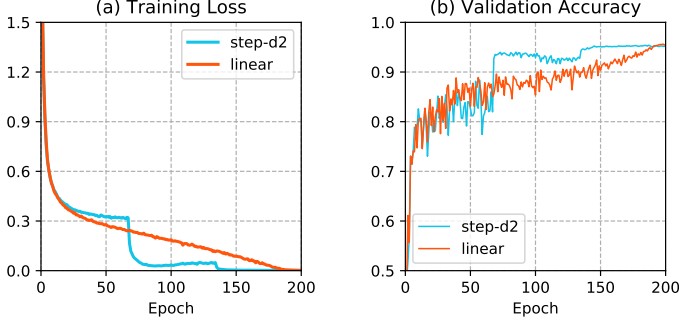

Figure C: Training loss and validation accuracy for training ResNet-18 on CIFAR-10 using step decay and linear schedule. No generalization gap is observed when we only modify learning rate schedule. This figure provides details for the discussion of "don't waste budget".

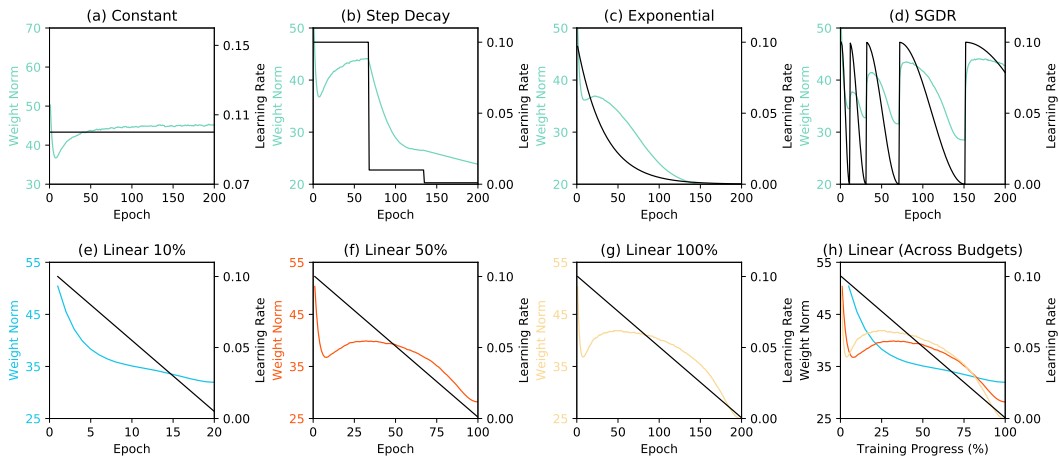

Figure D: The corresponding weight norm plots for Fig 3 and Fig 5. We find that the weight norm exhibits a similar trend as the gradient norm.

## I    FULL GRADIENT NORM AND THE WEIGHT NORM

In Sec 5, we plot the full gradient norm of the cross-entropy loss, excluding the regularization part. In fact, we use an L2-regularization (weight decay) of 0.0004 for these experiments. For completeness, we plot the weight norm in Fig D.

## J    ADDITIONAL ABLATION STUDIES

Here we explore variations of batch size (Tab H) and initial learning rate (Tab I). Our definition of budget is the number of examples seen during training. So when the batch size increases, the number of iterations decreases. For example, on CIFAR-10, the full budget is training with batch size 128 for 200 epochs. If we train with batch size 1024 for 20% of the budget, that means training for 5 epochs.

| Batch Size | Schedule | 20% | 50% | 100% |
|---|---|---|---|---|
| 64 | step-d2 | $.9436 \pm .0037$ | $.9505 \pm .0009$ | $.9519 \pm .0009$ |
| 64 | linear | $\mathbf{.9473} \pm .0021$ | $\mathbf{.9511} \pm .0008$ | $\mathbf{.9526} \pm .0020$ |
| 256 | step-d2 | $.8939 \pm .0027$ | $.9291 \pm .0021$ | $.9431 \pm .0008$ |
| 256 | linear | $\mathbf{.9143} \pm .0018$ | $\mathbf{.9415} \pm .0038$ | $\mathbf{.9484} \pm .0013$ |
| 1024 | step-d2 | $.5851 \pm .0460$ | $.7703 \pm .0121$ | $.8805 \pm .0007$ |
| 1024 | linear | $\mathbf{.7415} \pm .0141$ | $\mathbf{.8553} \pm .0023$ | $\mathbf{.8992} \pm .0042$ |

Table H: Comparison between linear and step decay with different batch sizes. We can see that even when we vary the batch size, linear schedule outperforms step decay.

| Initial LR | 0.001 | 0.1 | 1 | 10 |
|---|---|---|---|---|
| step-d2 | $.9152 \pm .0024$ | $.9529 \pm .0009$ | $.8869 \pm .0065$ | N/A |
| linear | $\mathbf{.9167} \pm .0023$ | $\mathbf{.9563} \pm .0009$ | $\mathbf{.8967} \pm .0034$ | N/A |

Table I: Comparison between linear and step decay with different initial learning rate under full budget setting. On one hand, we see that linear schedule outperforms step decay under various initial learning rate. On the other hand, we see that initial learning rate is still a very important hyper-parameter that needs to be tuned even with budget-aware, smooth-decaying schedules.

## K  ADDITIONAL IMPLEMENTATION DETAILS

**Image classification on ImageNet.** We adapt both the network architecture (ResNet-18) and the data loader from the open source PyTorch ImageNet example[5]. The base learning rate used is 0.1 and weight decay $5 \times 10^{-4}$. We train using 4 GPUs with asynchronous batch normalization and batch size 128.

**Video classification on Kinetics with I3D.** The 400-category version of the dataset is used in the evaluation. We use an open source codebase[6] that has training and data processing code publicly available. Note that the codebase implements a variant of standard I3D (Carreira & Zisserman, 2017) that has ResNet as the backbone. We follow the configuration of `run_i3d_baseline_300k_4gpu.sh`, which specifies a base learning rate 0.005 and a weight decay $10^{-4}$. Only learning rate schedule is modified in our experiments. We train using 4 GPUs with asynchronous batch normalization and batch size 32.

**Object detection and instance segmentation on MS COCO.** We use the open source implementation of Mask R-CNN[7], which is a PyTorch re-implementation of the official codebase Detectron in the Caffe 2 framework. We only modify the part of the code for learning rate schedule. The codebase sets base learning rate to 0.02 and weight decay $10^{-4}$. We train with 8 GPUs (batch size 16) and keep the built-in learning rate warm up mechanism, which is an implementation technique that increases learning rate for 0.5k iterations and is intended for stabilizing the initial phase of multi-GPU training (Goyal et al., 2017). The 0.5k iterations are kept fixed for all budgets and learning rate decay is applied to the rest of the training progress.

**Semantic segmentation on Cityscapes.** We adapt a PyTorch codebase obtained from correspondence with the authors of PSPNet. The base learning rate is set to 0.01 with weight decay $10^{-4}$. The training time augmentation includes random resize, crop, rotation, horizontal flip and Gaussian blur. We use patch-based testing time augmentation, which cuts the input image to patches of $713 \times 713$ and processes each patch independently and then tiles the patches to form a single output. For overlapped regions, the average logits of two patches are taken. We train using 4 GPUs with *synchronous* batch normalization and batch size 12.

---

[5] https://github.com/pytorch/examples/tree/master/imagenet. PyTorch version 0.4.1.

[6] https://github.com/facebookresearch/video-nonlocal-net. Caffe 2 version 0.8.1.

[7] https://github.com/roytseng-tw/Detectron.pytorch. PyTorch version 0.4.1.

