# OpenReview forum: "Budgeted Training: Rethinking Deep Neural Network Training Under Resource Constraints"
_ICLR.cc/2020/Conference — Accept (Poster)_

### Official Review · AnonReviewer3 · 2019-10-20
**Official Blind Review #3**

**Rating:** 6

**Review:**

This work presents a simple technique for tuning the learning rate for Neural Network training when under a "budget" -- the budget here is specified as a fixed number of epochs that is expected to be a small fraction of the total number of epochs required to achieve maximum accuracy. The main contribution of this paper is in showing that a simpler linear decay schedule that goes to zero at the end of the proposed budget achieves good performance. The paper proposes a framework called budget-aware schedule which represents any learning rate schedule where the ratio of learning rate at time `'t' base learning rate is only a function of the ratio of 't' to total budget 'T'. In this family of schedules, the paper shows that a simple linear decay works best for all budgets. In the appendix, the authors compare their proposed schedule with adaptive techniques and show that under a given budget, it outperforms latets adaptive techniques like adabound, amsgrad, etc.

Pros:
1. This paper presents a simple technique for a problem that is impactful namely performing training under a small budget presumably as an approximation during neural architecture search or hyperparameter tuning. The technique is empirically shown to be effective for many computer vision benchmarks.
2. The paper presents extensive experimental results comparing linear decay with other budget-aware schedules. The accuracy comparisons are performed under different budgets as well as for neural architecture ranking while selecting architecture with budgeted training.
3. Overall, I think this paper can be generally useful for many practitioners.

Cons:
1. The paper makes claims around the phenomena of gradient magnitude vanishing as well as its effectiveness. E.g. in section 5, authors state "We call this “vanishing gradient” phenomenon budgeted convergence. This correlation suggests that decaying schedules to near-zero rates (and using BAC) may be more effective than early stopping.". This is not clear from the paper as the paper merely shows gradient magnitude decreasing with learning rate. This claim appear like an overreach to me.
2. The key motivating use cases for budget-aware training is providing approximations for problems like neural architecture search and hyper parameter tuning. However, for these use cases, the paper does not perform extensive comparisons for commonly used algorithms like Adam. Why?

nits:
1. In section 2, various -> varies
2. Right above equation 1, budge -> budget


**Experience Assessment:**

I do not know much about this area.

**Review Assessment: Checking Correctness Of Derivations And Theory:**

I assessed the sensibility of the derivations and theory.

**Review Assessment: Checking Correctness Of Experiments:**

I assessed the sensibility of the experiments.

**Review Assessment: Thoroughness In Paper Reading:**

I read the paper thoroughly.

---

> ### Author Response · Authors · 2019-11-13
> **Response for R3**
>
> Thanks for reviewing our paper!
>
> 1. The effects of a “vanishing gradient” may be more obvious in Fig 5 (right), where a small loss is achieved only when the gradient magnitude is small. We will work on better explanations and improve the clarity here.
>
> 2. We agree that it would be more comprehensive to include algorithms like Adam. However, for most experiments, we focus on comparing budget-aware linear schedules with off-the-shelf defaults (which we assume have already been aggressively tuned in the state-of-the-art codebases for each of our benchmark tasks). For NAS, most of the papers we found adopt SGD. And for all practical vision benchmarks, momentum SGD is used. There is no particular reason not to include Adam except keeping our experiments to a manageable size. As we argue for R2, our evaluations are significantly larger scale than other papers in our peer group. Finally, we point out that we have included AMSGrad in Fig 4 & 6, which is designed to improve upon Adam itself.
>
> 3. We have fixed these typos. Thanks for pointing out!

---

### Official Review · AnonReviewer1 · 2019-10-23
**Official Blind Review #1**

**Rating:** 6

**Review:**

This paper analyzed which learning rate schedule (LRS) should be used when the budget (number of iteration) is limited. First, the authors have introduced the concept of BAS (Budget-Aware Schedule). Various LRSs are classified, and it is experimentally shown that the LRSs based on BAS performed better. Among them, the performance of the linear decay method was shown to be simple and robust.

Pros

1. Formally define important and well-motivated issues to improve performance on a limited budget.
2. Various experimental results show that the simple linear decay method works well, and it might become a baseline method for future budgeted training solutions (assuming there is no similar work with the same purpose).
3. It seems to be easy to apply to the NAS,  and the experiments in Appendix A look a big plus. This section can be put into the main paper.

Cons

1. No sound theory as to why linear decay or other smooth decay methods work well.
2. As Mishkin et al. [1] have already experimented with linear decay, the novelty of the methodology proposed by the authors might be limited.

While there is some concern regarding the significance of novelty,  the paper seems meaningful enough to be accepted.

[1] Dmytro Mishkin, Nikolay Sergievskiy, and Jiri Matas. Systematic evaluation of convolution neural network advances on the imagenet. Computer Vision and Image Understanding, 161: 11–19, 2017.

**Experience Assessment:**

I have read many papers in this area.

**Review Assessment: Checking Correctness Of Derivations And Theory:**

I assessed the sensibility of the derivations and theory.

**Review Assessment: Checking Correctness Of Experiments:**

I carefully checked the experiments.

**Review Assessment: Thoroughness In Paper Reading:**

I read the paper thoroughly.

---

> ### Author Response · Authors · 2019-11-13
> **Response for R1**
>
> Thanks for reviewing our paper!
>
> In the deep learning era, it might often be the case that practice precedes theory. We hope that our budgeted setting allows new optimization algorithms to appear. Many existing optimizers focus on the goal of decreasing the training loss as fast as possible, which is not required in the budgeted setting. We are also open to any suggestions on how to develop a sound theory to explain the findings in this paper.

---

### Official Review · AnonReviewer2 · 2019-11-07
**Official Blind Review #2**

**Rating:** 6

**Review:**

Pros:
The paper is clearly written. It provides an interesting perspective for training neural networks under resource constraints. The problem setting is novel. The proposed solution is simply decaying learning rate linearly from the initial value to zero during training, which is parameter free.

Cons:

- As the authors are advocating using linear scaling schedule, I would like to see whether it has some clear advantages over other schedules, but it is not quite clear. For example, we can still see step based schedule has better performance in 2 of the 4 tasks in Table 2. Poly and Cosine schedule is also better in some of the budgets in Figure 2.

- The comparison in Figure 2 and Table 2 is not very convincing without considering the variance of different trails. It is not clear whether the advantage is caused by learning rate schedule or randomness. It is better to report the mean and variance for multiple trials. Ideally, it would be better to performance significance test.

- I would like to see other lr schedules in Table 2. As shown in Figure 2, step based schedule is not the top3 schedules for CIFAR-10.

- As shown in Table 3, the proposed method has to wait until the end of training to get the best performing model, while step based schedule can find the best model around 90% training. The author argues that the proposed method does not need to perform validation test for each checkpoint and reduce the computation cost,  however, on the other side, this means early stopping is not able to use for linear scaling based schedule, which could be very useful when the training budget is large enough and evaluation is cheap.

- My major concern for this work is a lack of deeper understanding about the reason why linear LR schedule works better, if any. It would be stronger with such understandings. The authors try to provide an explanation from the relationship between learning rate and gradient magnitudes, but no clear conclusion is given. As noted in [1,2,3], when weight decay is used in training and BN layers are used, the weight magnitude is also decreasing, so is the gradient norms. But the weight norms or gradient norms does not mean too much due to scale invariance. I would like to see when no weight decay is used and whether there is any correlation between the learning rate and gradient norms.

- What is the lr decay unit for linear schedule? Is it decaying per epoch or per mini-batch? If epoch based lr decay is used, it is essentially step-based lr decay with many steps. Then when the number of epochs is three, the step decay method (lr decays at epoch 1 and 2) and the linear decay method are actually almost equivalent.

- I would expect the convergence to be related with number of iterations. When the number of iterations is not long enough, neither linear lr decay or step based decay will work. It would be better if the author can investigate when linear schedule starts to outperforms step based decay in terms of epochs or iterations. Since different batch size results in different number of iterations, I would expect the difference between two schedules for small batch size at the early stage of training would be less in comparison with large batch training, especially when the number of iterations is enough.

- Different initial learning rate may also results in different behaviour. We often see some learning curves with larger initial learning rate converges faster at the beginning but yields similar generalization error at the end of training. On the other hand, the author only compared different schedules with single initial learning rate. Image when the initial learning rate is small, there would not be too much difference for different schedules. Actually the linear decay schedule changes may simply find a good learning rate during training as long as the initial learning rate is larger than the optimal one.


[1] Dinh et al, Sharp minima can generalize for deep nets, ICML 2017
[2] Li et al, Visualizing the Loss Landscape of Neural Nets, NIPS 2018
[3] van Laarhoven, L2 regularization versus batch and weight normalization, NIPS 2017


----- update after rebuttal ------
The authors' rebuttal addressed some of my concerns.  I think early stopping is still an important feature to save compute, especially for HPO, which could limit the usage of the proposed method (cannot stop earlier). If the authors believe practitioners are used to train model in full budget without early stopping to guarantee the best performance, then why would people try budged training with worse performance? I hope the authors could make the limitation clear. My major concern about the reason for why linear scaling schedule is better is still not clear, which makes the contribution kind of weak. Nevertheless,  the problem setting and the observations could be beneficial to the community for further discussion, So I raised my score to weak accept.

minor: I see exactly the same variance values for step and linear methods int the top rows of Table 12, is this a mistake?


**Experience Assessment:**

I have published one or two papers in this area.

**Review Assessment: Checking Correctness Of Derivations And Theory:**

N/A

**Review Assessment: Checking Correctness Of Experiments:**

I carefully checked the experiments.

**Review Assessment: Thoroughness In Paper Reading:**

I read the paper thoroughly.

---

> ### Author Response · Authors · 2019-11-13
> **Response and additional experiments for R2**
>
> Thanks for reviewing our paper!
>
> As requested, we added mean and std tables (Table 11 and 12) in Appendix I. For reference, we previously reported the median in Fig 2 and Table 2. R2’s first concern is full-budget training: we find that the improvement of linear decay over step decay is sometimes not statistically significant (e.g., both are nearly identical for ImagetNet training with the full-budget). However, at budget-constraints settings (our focus! - e.g., than 50% iterations), linear definitely and consistently outperforms step-decay.
>
> The second concern regards the relative performance between linear, cosine and poly. Our primary empirical observation, illustrated in Fig 1, is that budget-aware, smoothly decaying schedules dramatically outperform other schedules in low-budget settings. We agree entirely that budget-aware poly and cosine curves, when plotted in Fig. 1, would look similar to linear. That is why we state in the second last paragraph on page 6:  “using cosine and poly will achieve similar performance as linear”. Linear schedules are not our primary contribution; rather budget-aware training is. That said, we choose to emphasize linear as the practical “go-to” budget-aware schedule because of its simplicity arising from the lack of parameters.
>
> We also agree with R2 that having other learning rate schedules in Table 2 would be more informative. But we simply ran out of resources to do so in the rebuttal period (each individual run takes 1 week on a 4 or 8 GPU machine). We emphasize that most papers in our peer group analyze performance on CIFAR, and perhaps ImageNet. Our analysis on multiple large-scale vision benchmarks is unique, and (in our opinion), makes our results particularly relevant. But it also makes it difficult to perform exhaustive ablations. Instead, we make the assumption that state-of-the-art codebases for such benchmarks are already well-tuned, and we “swap out” default learning-rates holding all else fixed. That said, we posit that cosine, poly, and linear will behave similarly, and all outperform step decay.
>
> It is true that our method cannot take advantage of early stopping. R2 is correct that, in theory, early stopping can be used to save computational resources. But our experience is that early stopping in practice, is not used to reduce compute (e.g., most practitioners still finish the full training run and apply early-stopping post-hoc for model selection to reduce overfitting).
>
> R3 points an alternative explanation to the decreasing gradient magnitude in Fig 3:  when weight decay and BN are used, the weight magnitude is also decreasing. We plotted more statistics in Appendix J to address this issue, including the requested no weight decay setting. First, for Fig 3-5, we plot the full gradient of the cross-entropy loss, excluding the regularization loss. For completeness, we plot the weight norm in Fig 9 (same setting with weight decay of 5e-4). We can see that there is a similar correlation between weight norm and learning rate. Note that here we swap out poly in Fig 3 with the SGDR in Fig 6 since poly exhibits similar trends as linear. To see the effect of weight decay, we re-run the experiments without weight decay and plot both the full gradient norm (of the cross-entropy loss only) and the weight norm in Fig 10 and Fig 11. We see the weight magnitude is increasing and there is still some though weaker correlation between the learning rate and the full gradient norm (most obviously in the SGDR plot). Therefore, our observation still holds and is consistent with the reference provided by R3. We hesitate in drawing a clear conclusion since we don’t want to over-claim. However, we do believe this “vanishing gradient” phenomenon is interesting and worth studying.
>
> In practice, the minimum budget we explore is 1 epoch. We find that linear outperforms step decay in this setting. R2 is correct that this may change for even smaller budgets, but such resource constraints seem artificially-limiting given the sizes of current datasets. For completeness, we added experiments of different batch sizes. The experiments suggest that R2’s hypothesis is correct: the gap between the two schedules is smaller with a small batch size than with a large one.
>
> We also added experiments about initial learning rates in Appendix K. We find the initial learning rate is a hyperparameter mostly orthogonal to the learning rate schedule studied in this paper. While our conclusion of linear schedule outperforming step decay still holds for different initial learning rates, the importance of initial learning rates still outweighs that of learning rate schedules. Specifically, It is not the case that the linear learning rate can work with larger initial learning rates.

---

> ### Author Response · Authors · 2020-01-26
> **Early Stopping and Variance in Table 12**
>
> We agree that the proposed method is not compatible with early stopping, which might limit its applicability to certain training schemes. Ideally, one would like to train for enough epochs and choose the best performing checkpoint. However, the training budget can be limited sometimes. As mentioned in the introduction, budgeted training is useful for future datasets which can be magnitudes larger, and for research development stage and neural architecture search (NAS) where only the relative performance is of interest. Our experiments suggest that in those cases, budget-aware schedules might be preferred over early stopping.  Though not directly related, hopefully, our NAS experiments in Appendix A address some of R2's concerns on HPO.
>
> Also, thanks for pointing out the mistake in Table 12 (now Table 3)! We have fixed it.

---

### Author Response · Authors · 2019-11-15
**Summary of changes for the revised version**

In the revision, the main text remains the same, and additional statistics and experiments are provided in the appendix. These additions are requested by the reviewers.

- We added mean and std tables (Table 11 and 12) in Appendix I for our main results (Fig 2 and Table 2). In the main text, we report the median of 3 runs.
- We added the weight norm plot in Fig 9 corresponding to Fig 3 (the gradient norm and the learning rate).
- We plotted Fig 3 & 9 in the setting with weight decay disabled in Fig 10 & 11.
- We added ablation studies about the batch size and the initial learning rate in Tab 13 & 14 respectively.

---

### Decision · Program_Chairs · 2019-12-19

**Decision:**

Accept (Poster)

**Comment:**

This paper formalizes the problem of training deep networks in the presence of a budget, expressed here as a maximum total number of optimization iterations, and evaluates various budget-aware learning schedules, finding simple linear decay to work well.

Post-discussion, the reviewers all felt that this was a good paper. There were some concerns about the lack of theoretical justification for linear decay, but these were overruled by the practical use of these papers to the community. Therefore I am recommending it be accepted.